# Recent increases in tropical cyclone intensification rates

Kieran T. Bhatia[1,2], Gabriel A. Vecchi [2,3], Thomas R. Knutson [1], Hiroyuki Murakami[1,4], James Kossin[5], Keith W. Dixon[1] & Carolyn E. Whitlock[1,6]

Tropical cyclones that rapidly intensify are typically associated with the highest forecast errors and cause a disproportionate amount of human and financial losses. Therefore, it is crucial to understand if, and why, there are observed upward trends in tropical cyclone intensification rates. Here, we utilize two observational datasets to calculate 24-hour wind speed changes over the period 1982–2009. We compare the observed trends to natural variability in bias-corrected, high-resolution, global coupled model experiments that accurately simulate the climatological distribution of tropical cyclone intensification. Both observed datasets show significant increases in tropical cyclone intensification rates in the Atlantic basin that are highly unusual compared to model-based estimates of internal climate variations. Our results suggest a detectable increase of Atlantic intensification rates with a positive contribution from anthropogenic forcing and reveal a need for more reliable data before detecting a robust trend at the global scale.

[1] NOAA/Geophysical Fluid Dynamics Laboratory, Princeton, NJ 08540, USA. [2] Geosciences Department, Princeton University, Princeton, NJ 08544, USA. [3] Princeton Environmental Institute, Princeton University, Princeton, NJ 08544, USA. [4] University Corporation for Atmospheric Research, Boulder, CO 80307, USA. [5] NOAA/National Centers for Environmental Information, Center for Weather and Climate, University of Wisconsin, Madison, WI 53706, USA. [6] Engility Inc., Dover, NJ 07806, USA. Correspondence and requests for materials should be addressed to K.T.B. (email: kbhatia@princeton.edu)

Society is faced with significant challenges when tropical cyclones (TCs) rapidly intensify. These storms quickly rise through the Saffir–Simpson intensity scale[1,2], occasionally jumping from a category 1 (64–82 knots) to category 5 (>137 knots) hurricane within a couple of days. Storms that undergo rapid intensification (RI; defined as the 95th percentile of 24-h intensity changes[3]) reach major hurricane status (wind speeds greater than 95 knots, categories 3–5 on the Saffir–Simpson Scale) about 80% of the time[4] and are associated with the highest forecast errors. As a result, RI can lead to disastrous scenarios when coastal areas are not given adequate notice to evacuate and prepare for an extremely intense TC (e.g., Hurricane Audrey, 1957)[5].

The destruction and forecast error associated with TCs that undergo RI has inspired new research on whether RI frequency will be altered due to climate change. A recent statistical-dynamical downscaling study found that the number of TCs that undergo RI before U.S. landfall is projected to significantly increase in the late 21st century compared to the late 20th century[5]. A separate study involving climate change simulations produced by a high-resolution global climate model (GCM) projected a dramatic increase in the global incidence of RI due to global warming[6]. The agreement in these two studies with different methodologies as well as the theoretical backing by a third study[7] suggests we should investigate whether climate change has already increased the probability of TCs rapidly intensifying. An anthropogenically forced signal in recent observational data would provide evidence that these model projections are well-grounded and therefore, substantial coastal mitigation and adaptation strategies would be required in the near future.

Although TC frequency has stayed approximately constant over recent decades, there is growing evidence that the proportion of TCs which become major hurricanes has significantly increased[8–13]. Therefore, intensification metrics that capture the relative, rather than the absolute, frequency of the highest intensification rates are useful for understanding whether the probability of RI events is increasing. For example, a recent study showed the 95th percentile of 24-h intensity changes significantly increased in the central and eastern tropical Atlantic basin during the period of 1986–2015[14]. Another intensification metric that is not dependent on TC frequency, the intensification rate of intensifying storms, exhibited significant growth between 1977 and 2013 in the West Pacific basin[15]. In both basins, the studies showed the large-scale environment becoming more conducive to TC intensification with time. Specifically, areas with the largest increase in sea surface temperatures (SSTs) and potential intensities[16] appear to be collocated with the largest positive changes in intensification rates.

Here, we build on these results by evaluating Atlantic and global TC intensification trends using different metrics and additional observational datasets. We focus primarily on the Atlantic basin because it is the basin with the most consistent high-quality observations. There are currently no published findings on global trends in TC intensity changes, so these results are also included to provide a more comprehensive perspective. To separate anthropogenic trends and normal climate fluctuations, we use a state-of-the-art coupled GCM, the high-resolution forecast-oriented low ocean resolution model (HiFLOR). Specifically, we simulate the year-to-year variations in TC intensification from a suite of HiFLOR multicentury experiments that serve as a proxy for natural climate variability and compare the magnitude of the trends in the simulated climate to those in observations . The recent increase in high-intensification rates in the Atlantic basin is outside the range of normal climate variability defined by HiFLOR, which suggests anthropogenic forcing increases the likelihood of TCs rapidly intensifying.

## Results

**Observational trends**. Thus far, TC intensification trend analysis has exclusively focused on International Best-Track Archive for Climate Stewardship[17] (IBTrACS) data (Methods), which likely adds systematic biases to published results. IBTrACS is composed of TC best-track data from different operational agencies whose TC intensity observations became more reliable when global satellite coverage was introduced in the early 1980s (except for the Indian Ocean which did not gain continuous satellite coverage until 1998)[18]. Over much of the globe, satellite data quality has improved through increasing resolution and decreasing view angles, which enhances the accuracy of the most recent intensity estimates. The recent addition of more in situ measurements in the Atlantic and the East Pacific leads to lower errors in these basins, which results in additional spatial and temporal inhomogeneities in the observations[19].

In order to gauge the impact of these inhomogeneities on TC intensification trends, we first compare the probability of 24-h wind speed changes in IBTrACS and a more homogeneous record of TC intensity, the Advanced Dvorak Technique-Hurricane Satellite-B1 (ADT-HURSAT)[12] (Methods). Kossin et al.[12] developed ADT-HURSAT in order to maintain the same protocol to determine TC intensities throughout all ocean basins during the period 1982–2009. ADT-HURSAT incorporates artificially degraded intensity estimates in certain regions so its data quality is consistent both spatially and temporally, which is better suited for trend analysis. However, this increased data consistency means the best technology and analysis techniques are not utilized in ADT-HURSAT, which leads to individual intensity estimates with higher average errors than those in IBTrACS[12,20]. However, for the scope of our study, which involves assessing rates of change throughout a time series, temporal consistency is preferable to accuracy in individual estimates.

Figure 1 shows the empirical probability density plots calculated from ADT-HURSAT and IBTrACS 24-h intensity changes, using a logarithm transformation on the probabilities to better illustrate differences in the tails of the distributions. Supplementary Figure 1 shows the more traditional probability density plots. For all of our analyses, we impose criteria involving storm longevity, location, and intensity to ensure only intensity changes derived from well-defined storms are included in our results (Methods). For the global results, we entirely exclude the north and south Indian Oceans from the sample because of the well-known absence of quality satellite data in the region until 1998[12]. Nevertheless, our conclusions do not significantly change when these basins are introduced into the analysis.

As expected, the differences in the ADT-HURSAT and IBTrACS intensity change distributions are larger for the global data, while the two datasets show more agreement in the Atlantic basin. For the global results, TCs in ADT-HURSAT maintain their intensity for a larger percentage of cases than in IBTrACS but also have significantly more instances of intensification rates between 50 and 90 knots (resulting in a "shelf" in this part of the distribution). IBTrACS curves appear almost linear in both plots which imply that probability densities are exponentially distributed. Thus, IBTrACS suggests that TC intensification and dissipation are likely controlled by statistically random environmental and internal processes[21]. Meanwhile, ADT-HURSAT's shelf-like pattern suggests that RI is a special process that could preferentially augment the probability of the highest intensification rates. The former interpretation is much more likely considering the ADT-HURSAT algorithm exhibits well-documented problems during TC eye formation (Methods).

As discussed in Bhatia et al.[6], the discrepancies among basins and datasets cannot be attributed to specific physical processes or data deficiencies. A summary of the potential explanations for

this behavior are included in the Methods section. We here argue that the agreement among the observational datasets in the Atlantic basin provides some level of confidence in the trend analysis in this basin. On the other hand, both observational datasets have larger uncertainties in global intensity changes. Therefore, any conclusions stemming from the global trend analysis must be treated with caution.

Figure 2 shows Atlantic basin and global trends in 24-h intensity changes between 1982 and 2009. The 28-year period is selected for analysis because it is the longest continuous sample where ADT-HURSAT produces relatively reliable global intensity data. Quantile regression is calculated for every 5% quantile between 5% and 95% (Methods). IBTrACS shows significant trends in all quantiles except the ones between 35% and 45%. The 95th percentile shows the largest trend of about +4 knots decade$^{-1}$. The negative slopes for the lower percentiles and positive slopes for the upper quantiles suggests a broadening of the intensity change distribution, implying less TCs maintaining a steady intensity and more TCs exhibiting high-intensity fluctuations. ADT-HURSAT shares a similar pattern for the slopes of the

quantiles but the trends are more muted. Despite being relatively small in magnitude, ADT-HURSAT quantile slopes greater than the 50% quantile are statistically significant and positive, with the 90th percentile having the largest slope of approximately 1 knot decade$^{-1}$.

For the global data, ADT-HURSAT trends are clearly much smaller than those in IBTrACS, yet IBTrACS trends are likely more susceptible to data homogeneity problems. Therefore, it is our interpretation that the weaker global trends shown by ADT-HURSAT are more likely to be correct. This conclusion is particularly notable because multiple studies[4,13–15] have assumed that IBTrACS is reliable during the time frame considered in our study when in reality, there is likely a spurious trend masked by the data. It is possible that resolution and algorithm issues with ADT-HURSAT prevent it from resolving the intensity changes and have slightly dampened largest trends. However, it is much more likely that the influx of new satellites has enabled better detection of intensity changes in the later years of the time series and thus, artificially enhanced the slope of the extreme quantiles in IBTrACS.

In the Atlantic basin, the quantile regression for ADT-HURSAT and IBTrACS yields similar results for extreme intensity changes. For both observational datasets, the 95th percentile shows the largest trend. This quantile increases by over +4 knots decade$^{-1}$ in ADT-HURSAT and +3 knots decade$^{-1}$ in IBTrACS. The upward trend in TC activity during the 1990s is typically attributed to the Atlantic Multidecadal Oscillation (AMO), which features multidecadal fluctuations in SSTs in the Atlantic basin[22]. In the late 1990s, the phase of the AMO shifted from negative to positive which coincides with warming in parts of the Atlantic basin where TCs develop[14]. Therefore, the AMO phase favored RI at the end of the time series. In addition to the AMO behavior, the reliability of the observational data in the Atlantic basin suggests the positive trend in intensification rates during 1982–2009 is likely robust. IBTrACS has its lowest errors in the Atlantic basin because of the plentiful in situ observations that are incorporated into the best-track analysis. ADT-HURSAT relies on best-track positions to accurately assign a scene type and find an eye, which suggests that the additional reconnaissance in this basin could also help with ADT-HURSAT's intensity estimates.

As a complement to quantile regression, we also analyze how the annual rapid intensification ratio (RI ratio) has changed from 1982 to 2009. RI ratio is defined as the number of 24-h intensity changes greater than 30 knots divided by the total number of intensity changes[6], and it reflects changes in the probability of the highest TC intensification rates. Figure 3 illustrates the Atlantic basin and global trends in RI ratio for ADT-HURSAT and IBTrACS. For both observational datasets, slopes are positive and significant in the global and Atlantic basin data. Much like Fig. 2, the Atlantic basin trends for the observational datasets are very similar. Over the 28-year period, the percentage of 24-h intensity changes that exceed an intensification rate of 30 knots approximately triples in IBTrACS and ADT-HURSAT. IBTrACS shows a similar rise in RI ratio for the global data but ADT-HURSAT has a much smaller slope.

Supplementary Figure 2 shows the annual RI ratio of the two observational datasets plotted against each other. In the Atlantic basin, 35.7% of the variance in IBTrACS RI ratio is explained by ADT-HURSAT RI ratio. For the global data, the percentage explained drops to 1.8%, further highlighting the lack of agreement between the two observational datasets at global scales. The variations in the slope of RI ratio among the global observations as well as the lack of a published theory[23] that explains the source of a significant positive slope in global TC intensification metrics highlight the uncertainty in the global

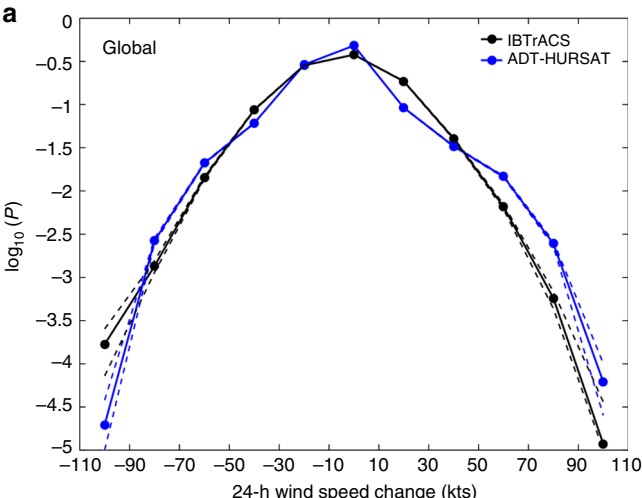

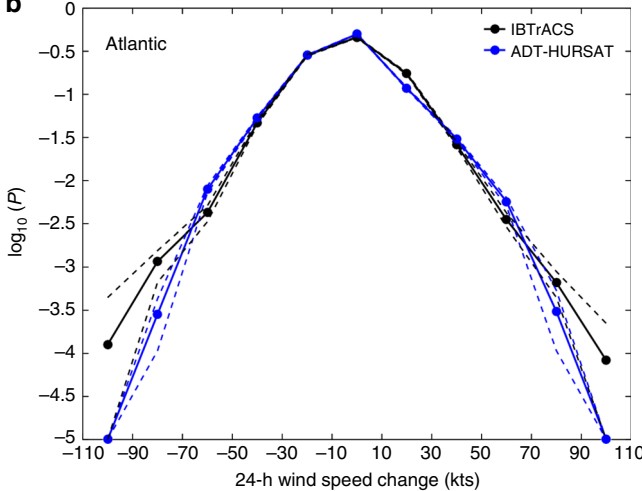

**Fig. 1** Observed 24-intensity change probability densities. **a**, **b** Common logarithm of the probability densities calculated from IBTrACS (black) and ADT-HURSAT (blue) 24-h intensity changes. **a** Global and **b** Atlantic basin results for the period 1982–2009 are plotted. Data are binned in 20 knot increments between −110 and 110 knots. The dashed lines indicate the 90% confidence interval (between 5th and 95th percentiles of the data) (Methods). All distributions are bounded below by 10$^{-5}$

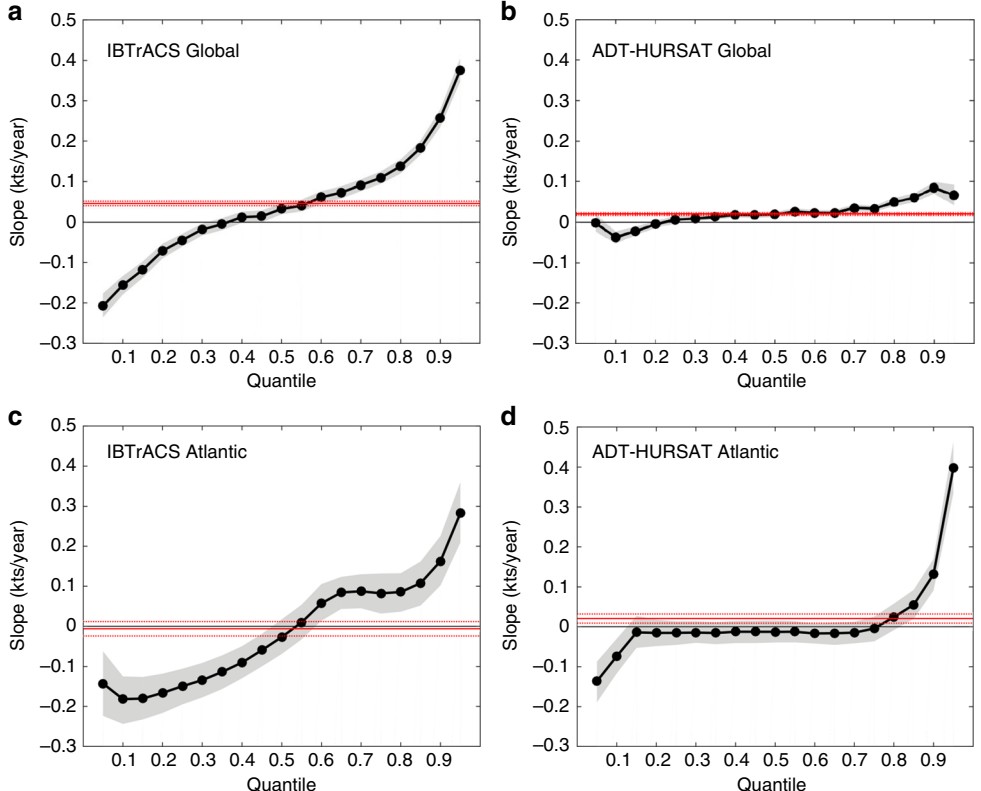

**Fig. 2** Quantile regression of 24-h intensity changes. **a–d** Slope of the quantiles for 24-h intensity changes during the period 1982–2009. Slopes are shown for IBTrACS (**a**, **c**) and ADT-HURSAT (**b**, **d**) globally (**a**, **b**) and in the Atlantic basin (**c**, **d**). The black dots represent the slope derived from least squares regression of intensity change as a function of year for each quantile from 0.05 to 0.95 in steps of 0.05. Shading represents the 5th and 95th percentiles of the regressions with randomly perturbed observational data (Methods). The red solid line shows the (constant value) trend in the mean as measured by ordinary least squares regression, and the red dotted lines show the 90% confidence interval

results. Conversely, the recent uptick in intense TCs and RI magnitude in the Atlantic basin appears more plausible because of the AMO shifting phase.

Ordinary trend analysis over a relatively short period (1982–2009) may mistake multidecadal natural variability for a long-term trend. For a more refined estimate of the possible influence of natural variability on observed trends and to quantitatively assess whether the recent increase in Atlantic RI ratio can be explained by natural variability alone, we compared the observed trends to those in a TC-permitting GCM that can simulate multidecadal climate variability in the Atlantic basin. Global results are also presented with the caveat that the ADT-HURSAT data are likely more realistic because of the homogeneous time series.

**Climate model simulations**. HiFLOR is a high-resolution coupled GCM that can recover many aspects of the highest TC intensification rates observed in nature and capture the connection between low-frequency climate oscillations and TC behavior[6,24–26]. A recent study[24] showed that HiFLOR, when nudged toward observed SSTs, can skillfully reproduce the observed year-by-year variations of the frequency of Category 4 and 5 hurricanes (maximum wind speed ≥ 113 kts) in the Atlantic basin ($r \approx 0.64$) and other basins to a lesser extent. HiFLOR's seasonal predictions of major hurricanes were also skillful in the Atlantic basin, and the high positive correlations between HiFLOR predictions and observations are still among the highest documented for a dynamical climate model[25]. The confirmed presence of important modes of internal climate variability within

HiFLOR as well as the resolved relationship between TCs and the large-scale climate instills some confidence that HiFLOR can diagnose whether observed changes are unusual compared to expected natural variability.

For an initial exploration of whether anthropogenic climate may be contributing to the increase in TC intensification, we follow the methodology of Murakami et al.[27] and examine TC variations from a suite of HiFLOR multicentury experiments with different levels of anthropogenic forcing. Four HiFLOR experiments are run using anthropogenic forcing (e.g., $CO_2$, aerosols, and ozone) and natural forcing (e.g., volcanic aerosol loading and solar insolation) representative of the years 1860, 1940, 1990, and 2015 (1860CTL, 1940CTL, 1990CTL, and 2015CTL; Methods).

To compare the control simulations to observations, Supplementary Figure 3 shows the logarithm of the 24-h intensity change probabilities observed between 1982 and 1998 in IBTrACS and ADT-HURSAT and those simulated in the 1990CTL. As a technique to account for the typical error associated with measuring TC intensity, the probability of each bin was calculated using 1000 subsamples that incorporated random error (Methods). To verify the quality of the control simulations, the 17-year period between 1982 and 1998 was selected for analysis because it is the longest available period with observational data that is centered on a control simulation. The 1990CTL data matches the shape of the IBTrACS curve but the distribution is not as broad as either of the observational datasets, indicating that HiFLOR underestimates the frequency of rapid weakening and intensification. A discrepancy in the probability of the highest intensity changes was also documented by Bhatia et al.[6] but the differences appear larger in Supplementary Figure 3,

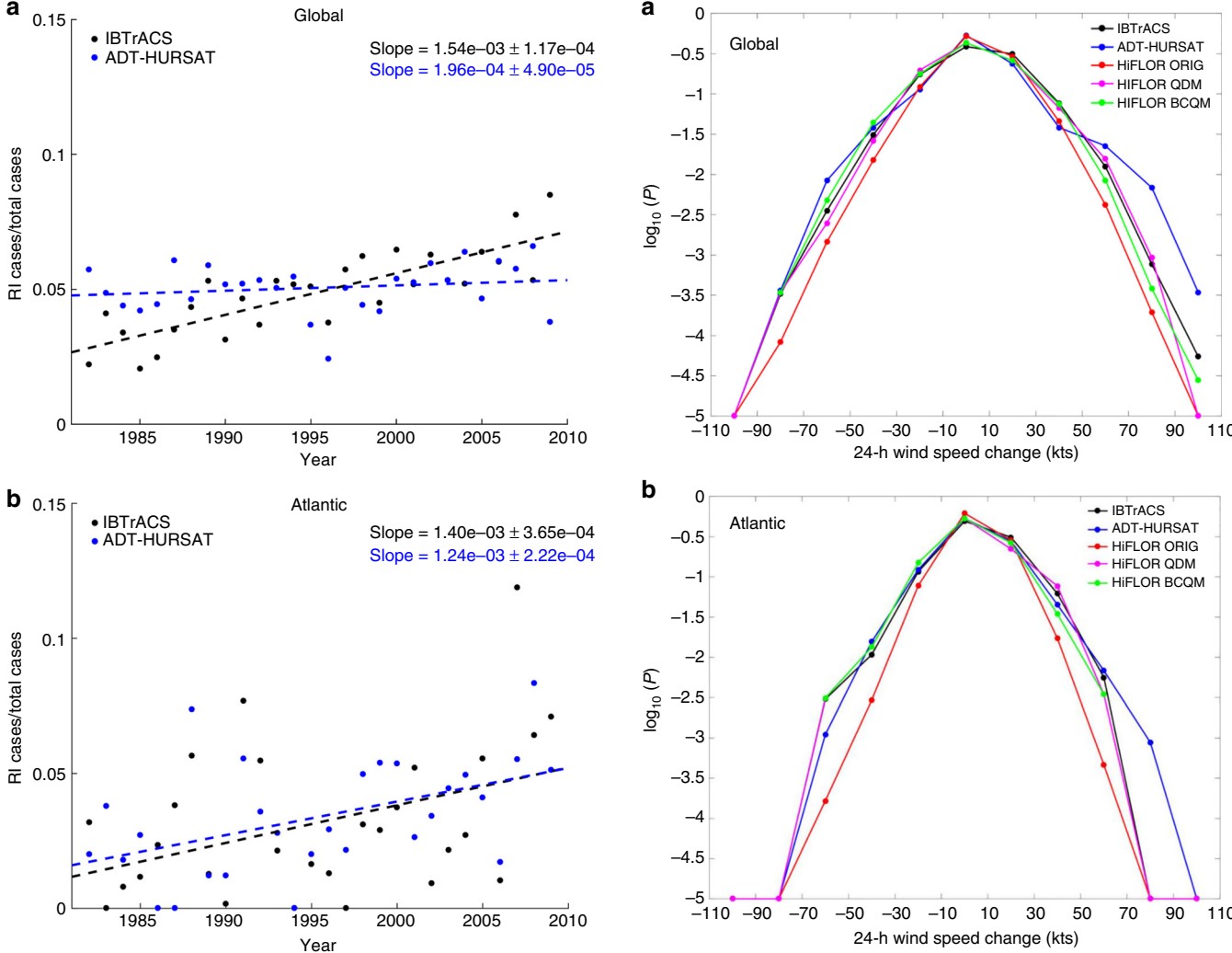

**Fig. 3** Rapid intensification ratio trends. **a**, **b** Observed trends in the rapid intensification (RI) ratio of ADT-HURSAT (black) and IBTrACS (blue) over the 28-year period 1982–2009 using **a** global and **b** Atlantic data. RI ratio is defined as the number of 24-h intensity changes above 30 knots divided by the total number of 24-h intensity changes. Trends in the time series of the annual mean RI ratio are denoted by dashed lines. The slopes of the trend lines as well as their 90% confidence intervals are provided. The slopes and confidence intervals are calculated using 1000 randomly perturbed samples of the observational data. Shading represents the 5th and 95th percentiles of the 1000 regressions with these randomly perturbed observational data (Methods)

**Fig. 4** The effects of quantile mapping on 24-intensity change probability densities. **a**, **b** Common logarithm of the probability densities calculated from IBTrACS (black), ADT-HURSAT (blue), HiFLOR 1990CTL (ORIG), HiFLOR QDM-corrected 1990CTL, and HiFLOR BCQM-corrected 1990CTL 24-h intensity changes. **a** Global and **b** Atlantic basin results are plotted. QDM and BCQM are quantile mapping designed to bias correct 1990CTL to more closely represent the distribution of IBTrACS. ADT-HURSAT and IBTrACS probabilities are computed using the 17-year period centered around 1990 (1982–1998). HiFLOR probability density curves are generated using 250 years of intensity changes from the 1990CTL. Data is binned in 20 knot increments between −110 and 110 knots, and each bin entry is plotted as a dot on a curve. All distributions are bounded below by $10^{-5}$

possibly due to the lack of SST nudging or the different years sampled.

HiFLOR underestimates the climatological rates of intensification, which will impact estimates of the amplitude of internal variability from this model. In order to mitigate the effects of systematic biases in HiFLOR intensity changes on estimating the internal variability of intensification rates, we utilize a statistical downscaling technique known as quantile mapping[28]. Specifically, we test two methods: quantile delta mapping[29] (QDM) and bias correction quantile mapping[30] (BCQM) (Methods). All subsequent analysis only incorporates the QDM algorithm because it is more adept at preserving relative changes in distributions of meteorological variables from different climate change simulations[29]. Figure 4 shows the effects of both quantile mapping algorithms. In this case, random error is not included in

the probability estimates because we aim to nudge HiFLOR to the best guess of the intensity distribution. Clearly, the bias corrections are aligning the tails of the 1990CTL more closely with the tails of the observations. When we explore the potential influence of anthropogenic forcing on TC intensification rates, it is critical that HiFLOR provides a realistic picture of natural climate variability. Bias-corrections increase the probability of extreme intensity changes in the HiFLOR control simulations, which broadens the intensity change distribution. Thus, the bias-corrections enhance the annual variations in RI ratio and the range of the RI ratio slopes in the HiFLOR simulations, which translates into more stringent statistical tests for establishing significant observational trends.

To test for observed RI trends that are outside of natural, internal climate variability, overlapping (not independent)

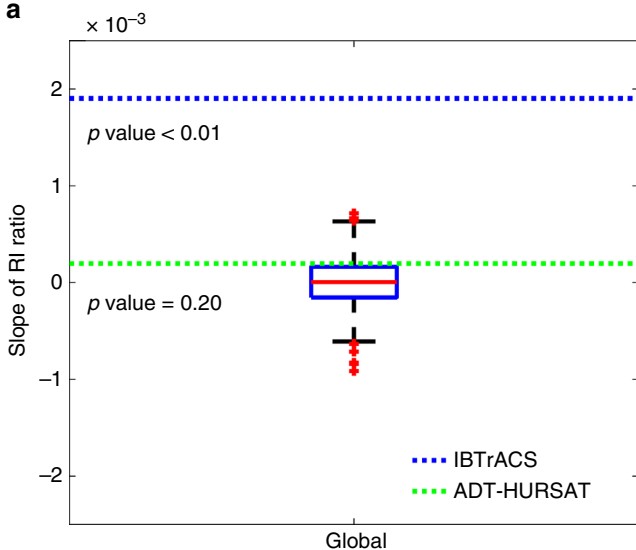

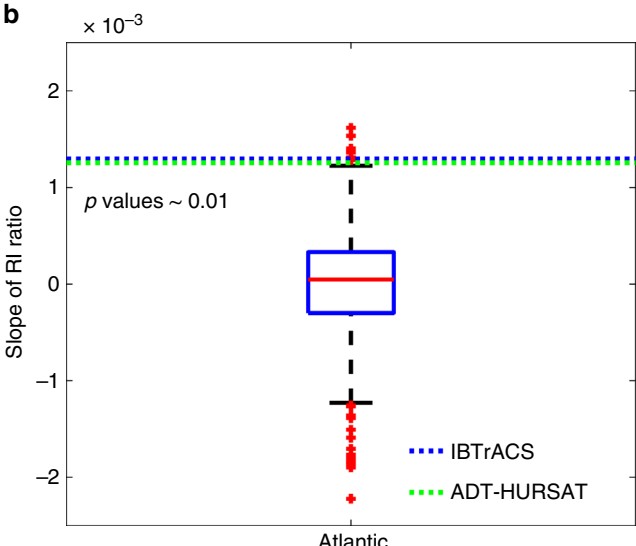

**Fig. 5** Observed trends in RI ratio vs. 1860CTL natural variability. **a**, **b** Box and whisker plot represents the distribution of slopes of RI ratio in the QDM-corrected 1860 HiFLOR control simulation. **a** Global and **b** Atlantic basin results are plotted. Each slope is calculated by applying least squares regression analysis to annual RI ratio values in overlapping 28-year periods. Thus, the number of slopes for a control simulation is the number of available years subtracted by 28 (i.e., 1860CTL has 1422 slopes). The red line in each box indicates the median of the slopes. The box is bounded by the 25th and 75th percentiles of the data, and the whiskers bracket approximately 99% of the data. Red plus signs indicate outliers whose values are outside of whiskers' range. IBTrACS and ADT-HURSAT trends in annual mean RI ratio between 1982 and 2009 are respectively represented by blue and green dotted lines and the corresponding p values are listed below each line

28-year RI ratio slopes in the bias-corrected HiFLOR control simulations are compared to the ADT-HURSAT and IBTrACS RI ratio slopes between 1982 and 2009. Supplementary Figure 4 shows how the probability distribution of RI ratio slopes changes when selecting 28-year segments from HiFLOR 1860CTL that are independent and not independent. The plotted bins show the independent sample has a slightly broader distribution but in general, the probabilities of the bins are similar to those in the overlapping bins.

Figure 5 contains a box and whisker plot that shows the distribution of RI ratio slopes for the QDM-corrected 1860CTL. The observed slopes during 1982–2009 are overlaid on each plot. Supplementary Figure 5 is similar to Fig. 5 but shows the box and whisker plots for all of the HiFLOR control simulations. Supplementary Figure 6 portrays the corresponding cumulative distribution functions (CDF) of the RI ratio slopes for all of the control simulations as well as the observed RI ratio slopes in ADT-HURSAT and IBTrACS. In the Atlantic basin, the slope of the RI ratio for ADT-HURSAT and IBTrACS are both above the 99th percentile of the slopes of any of the bias-corrected HiFLOR control simulations. Therefore, the large positive slope of RI ratio in both observational datasets is outside HiFLOR's estimate of expected internal climate variability, which suggests the model's depiction of climate oscillations like the AMO cannot explain the observed trend.

For the global data, IBTrACS data indicates a trend that is well above the bounds of the observed natural variability of RI ratio in any HiFLOR control simulation, but the noted temporal and spatial heterogeneities leads us to question the validity of this result. The slope of RI ratio for ADT-HURSAT is approximately the 80th percentile of the HiFLOR 1860CTL, indicating the observed trend is not significant compared to modelled climate variability. Importantly, ADT-HURSAT observed data also yields an RI slope that is still relatively unusual compared to the HiFLOR preindustrial natural variability. This emerging trend is consistent with an increasing trend in the global intensity of TCs[12]. In the near future, ADT-HURSAT will be extended to 2016, and it is therefore vital for future analysis to determine whether the global trend also becomes significant.

In order to explore whether the observed changes between 1982 and 2009 could be attributable to anthropogenic forcing, we compare RI ratio in the three climate change simulations to the preindustrial 1860CTL. Figure 6 shows the percent difference in RI ratio between 1860CTL and the HiFLOR simulations with stronger anthropogenic forcing (1940CTL, 1990CTL, and 2015CTL). The analysis only involves HiFLOR simulations so bias corrections and random error are not applied to the data (Methods). Grid boxes that are not statistically significant are demarcated with a white "X". The prevalence of red boxes throughout the maps conveys that a larger percentage of TCs are undergoing RI in 1940CTL, 1990CTL, and 2015CTL than in 1860CTL. There are no significant blue grid boxes in the 1990CTL and 2015CTL maps, and the number of gridboxes that record significant increases in RI ratio is 41 (9.7% of all shaded gridboxes) in 1940CTL, 156 (34.7% of all shaded gridboxes) in 1990 CTL, and 164 (38.2% of all shaded gridboxes) in 2015CTL. Historical radiative forcing changes applied to HiFLOR clearly lead to more frequent large intensification rates, but the available simulations do not allow us to strictly attribute the identified growth in RI during the period 1982–2009 to anthropogenic forcing. An ensemble of experiments that simulate anthropogenically forced climate change during this specific period, including changes in RI ratio, would be better suited for more comprehensive attribution analysis. Thus, for now, we can only conclude that anthropogenic forcing significantly increases extreme TC intensification rates in the HiFLOR model compared to preindustrial (1860CTL) conditions.

In the two most reliable long-term observational records available for TC intensity changes, the proportion of 24-h TC intensification rates greater than 30 knots significantly increases in the Atlantic basin between 1982 and 2009. Results averaged over all basins show a significant increase in TC intensification rates in IBTrACS but not in ADT-HURSAT. By itself, a 28-year upward trend in a TC intensification metric does not necessarily reflect the effects of anthropogenic climate forcing because of the

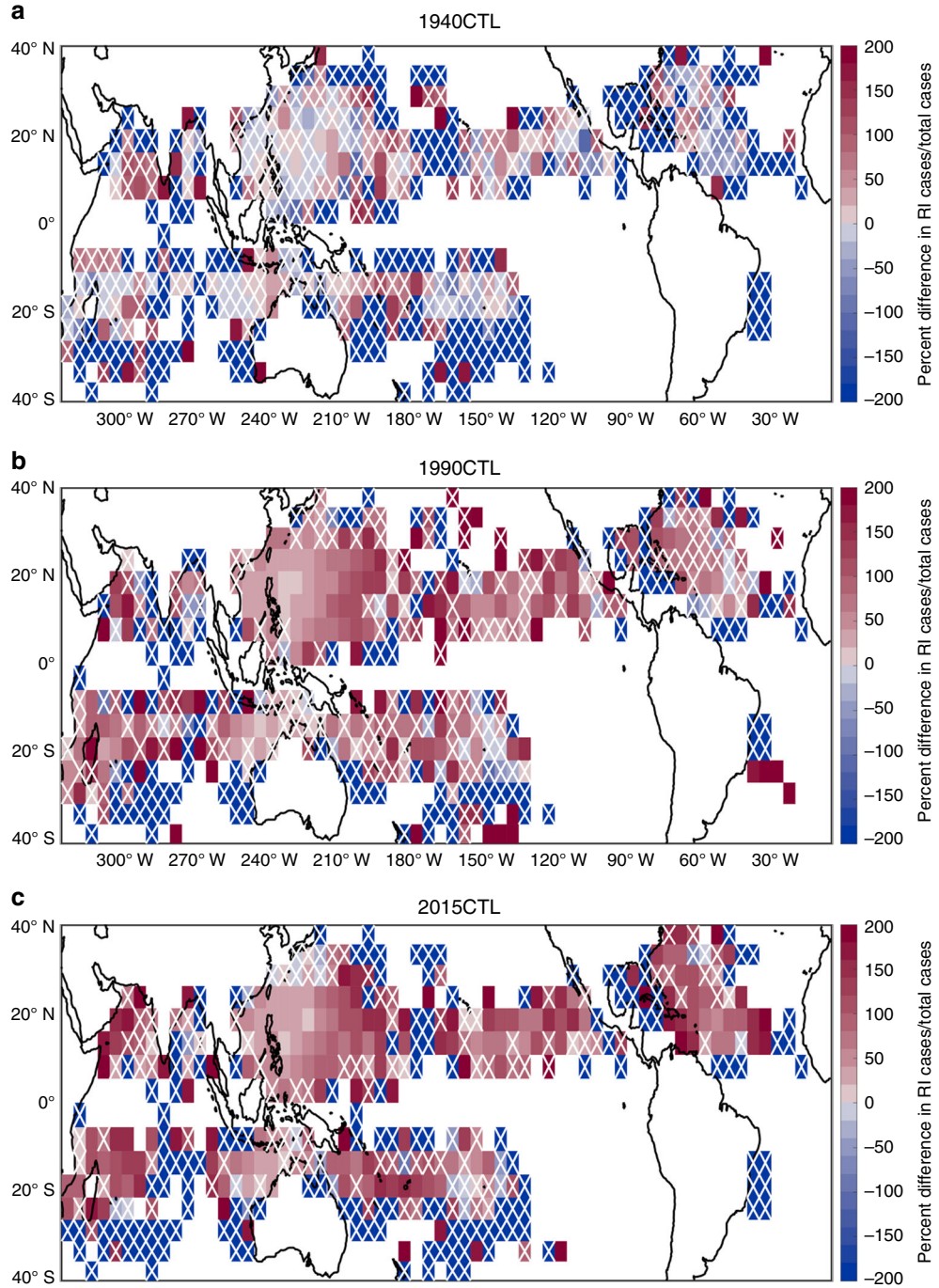

**Fig. 6** Anthropogenic forcing's effects on RI ratio in HiFLOR. **a–c** Simulated changes in RI ratio by the 1940CTL (**a**), 1990CTL (**b**), and 2015CTL (**c**) relative to the 1860CTL. Percent difference in RI ratio between HiFLOR 1860CTL and each climate change simulation is plotted in each 5° × 5° grid box. Data is only plotted in a grid box if at least one TC passes through the grid box every 50 years in the two experiments used to calculate percent difference. Red (blue) squares indicate grid boxes where a larger (smaller) percentage of 24-h intensity changes exceed 30 knots in the climate change simulations than in the 1860CTL. Grid boxes that achieve a *p* value of 0.05 using a binomial proportion test are considered statistically significant. White "Xs" are located in grid boxes that are not statistically significant

intrinsic natural variability of the climate system. Here, we use bias-corrected TC intensification rates simulated by HiFLOR to demonstrate that natural variability cannot explain the magnitude of the observed upward trend in the Atlantic basin. These conclusions are possible because HiFLOR is a unique climate model that can successfully simulate the most intense TCs and highest intensification rates in multicentury simulations.

This study is limited by the ability of a climate model to accurately represent natural variability as well as the uncertainty around the trends in relatively short observational records. However, this study represents a crucial first step in quantifying the precise roles of stochastic processes, anthropogenic warming, and natural variability when assessing changes in TC intensifica-tion rates. Further analysis with additional high-resolution

climate models and a longer and more reliable observational record is required to confirm these conclusions. Regardless, these trends provide another example of the potentially serious repercussions of anthropogenic warming for TCs.

## Methods

**Observed data**. We used the International Best Track Archive for Climate Stewardship (IBTrACS)[17], v03r09, and the Advance Dvorak Technique-Hurricane Satellite-B1[12] (ADT-HURSAT) for the period 1982–2009. For our IBTrACS analysis, we only consider best-track data from the National Hurricane Center for the Atlantic and east Pacific and the Joint Typhoon Warning Center for the remainder of the globe[17]. One of the benefits of only using data from these U.S. agencies is they follow the same definition of maximum winds: the highest 1-min average at 10 m height over a smooth surface[31]. Best-track data start as operational estimates of the intensity and track of a TC and are refined at the end of a TC's lifetime with a combination of in situ (e.g., dropsondes, scatterometers, and buoys), radar, and satellite measurements. Best-track intensity and position estimates are available every 6 h at the four synoptic times (0000, 0600, 1200, and 1800 UTC) and are recorded to the nearest 5 knots (1 kt = 0.5144 m s$^{-1}$) and 0.1° latitude/longitude[32].

The creation of ADT-HURSAT consists of four main steps. Geostationary satellite imagery is first analyzed from International Satellite Cloud Climatology Project (ISCCP)-B1 data[33–35]. Then, the data are centered on IBTrACS TCs and subsampled to be both spatially and temporally homogeneous. Finally, a simplified version of the advanced Dvorak technique[20] is used to evaluate the data and determine a maximum TC wind speed. ADT-HURSAT data are produced every 3 h based on satellite data that has been uniformly subsampled to a horizontal resolution of 8 km, and wind speeds are recorded to the nearest tenth of a Dvorak "T-number" (depending on the current intensity, between 1 and 3 knots).

Although ADT-HURSAT and IBTrACS represent the two most reliable multi-decadal observational datasets for TC intensification trend analysis, they both have limitations. In IBTrACS, Atlantic and East Pacific in situ observations have rapidly improved in the last 30 years, while other basins were rarely able to obtain aircraft reconnaissance. In fact, the west Pacific was the only other basin that had access to aircraft reconnaissance but these observations stopped in 1987. As a result, the number of measurements available in different basins varies considerably, leading to temporal and spatial inconsistencies in IBTrACS observational quality. Better intensity estimates at the end of the historical record potentially causes heightened detection of the larger intensity changes. Additionally, best-track estimates are also susceptible to human error because they are assembled at operational centers.

Compared to IBTrACS, ADT-HURSAT employs reduced-resolution satellite imagery, both in space and time. The omission of in situ measurements to accompany the coarsened satellite imagery is especially problematic for ADT-HURSAT identifying "scene type" changes when an eye is forming. As a result, there is a well-documented unphysical plateau when the eye formation stage of TC development in intensity observations from ADT-HURSAT. Low intensity changes are present for long periods of time and then the ADT-HURSAT is forced "to catch up" with higher intensity changes once it finally detects an eye. Thus, peak TC intensities (and changes) may not be fully resolved by ADT-HURSAT, especially in the cases of a transient, small eye. However, these deficiencies remain consistent throughout the time series, which suggests that there is a low likelihood of a systematic bias in ADT-HURSAT trends. It is also possible that larger eyes and a longer mean duration of the "plateau" period for TCs in the Atlantic basin could lead to minimal effects of these deficiencies.

**Criteria for inclusion in sample**. For consistency, intensity change values in HiFLOR and the observational datasets are rounded to the nearest five knots. We only consider TCs that are active for at least 72 h and exceed wind speeds of 34 knots for at least 36 h. We restrict our analysis sample to only consider cases where the TC center is located over the ocean, the starting and ending TC position are below 40° of latitude, and the TC intensity stays above 34 knots. The warm core criteria discussed in Murakami et al.[24] is also applied to the HiFLOR data before analysis.

**Control experiments**. Four HiFLOR control simulations introduced in Murakami et al.[27] were used here to represent natural climate variability and provide the framework for exploring anthropogenic effects. Control simulations were created using anthropogenic forcing fixed at 1860 (1860CTL), 1940 (1940CTL), 1990 (1990CTL), and 2015 (2015CTL) levels. Owing to limited computational resources, 1860CTL, 1940CTL, 1990CTL, and 2015CTL were run for different lengths: 1500, 200, 300, and 200 years, respectively. The first 50 years of all simulations were disregarded to mitigate effects of model drift. Basic conclusions remained similar even when we using the smallest sample size (150 years) for all the control simulations. The fixed forcing agents for the control simulations were atmospheric $CO_2$, $CH_4$, $N_2O$, halons, tropospheric and stratospheric $O_3$, anthropogenic tropospheric sulfates, black and organic carbon, and solar irradiance.

**Uncertainty quantification**. ADT-HURSAT and IBTrACS respectively provide intensity estimates to the nearest 1–3 and 5 knots. We use Monte Carlo techniques to create random noise before analyzing the discretized data. Random noise prevents multiple data points from having the same value and provides an estimate of the typical error associated with measuring TC intensity. 1000 subsamples were produced by adding random noise from a uniform distribution on the interval $\pm 2 \times \sqrt{50}$ knots to each intensity change value. The magnitude of this random noise is derived by adding 5 knots of error in quadrature (propagation of errors stemming from the intensity change calculation), which is a conservative estimate for the typical error associated with each TC intensity observation[35,36]. To calculate the slope of any intensification metric for ADT-HURSAT and IBTrACS, we use the slope of the mean of the 1000 subsamples for that particular metric. For example, quantile regression involved the calculation of every 5th percentile in each of the 1000 subsamples for each year. One-thousand slopes of each percentile were calculated and the mean of the slopes was considered the best estimate of the 1982–2009 slope of the quantile. The 5th and 95th percentiles of the 1000 slopes were considered the uncertainty bounds for the quantile.

For the creation of Fig. 6, random error is not added to the data because we only compare spatial differences in RI ratio among the different HiFLOR simulations. Statistical significance is computed using a binomial proportion test with $p$ values below 0.05 considered significant[37]. Data is only plotted in a grid box if there is at least one-fourth of a TC day per year in the two HiFLOR simulations used to calculate the percent difference.

**Quantile mapping**. In this work, two univariate bias correction methods were tested. We primarily focus on results produced using an additive version of QDM[29] by making use of R programming language code contained in the CRAN MBC package version 0.10–4[38]. As reported in Appendix A of Cannon et al.[29], the additive version of QDM is functionally very similar to the equidistant CDF matching algorithm of Li et al.[39]. The formulation of QDM aims to preserve relative changes in model-simulated climate variable quantiles. In other words, with respect to the quantiles, QDM guards against distorting the underlying climate model's climate sensitivity. For comparison purposes, we also used a simpler and more standard quantile mapping approach called BCQM. The more commonly used standard BCQM method develops transfer equations solely from comparisons of observations and model data sets drawn from a common historical (baseline) period, which can yield results in which the relative trends in bias corrected output differ from those of the raw GCM results. Meanwhile, the QDM method also incorporates adjustments accounting for differences between the modeled distributions of the historical period and future projections. As is generally the case for statistical bias correction methods, QDM and BCQM both assume that the GCM biases present in the historical period are consistent with biases in the GCM's future projections (i.e., a stationarity assumption).

For both of these mappings, IBTrACS serves as the training dataset that 1990CTL data is nudged to because its observations have lower errors than ADT-HURSAT. ADT-HURSAT was also tested as the target distribution for the bias correction, but these results are excluded since the basic conclusions of the results remain the same. After establishing the transfer function for the bias correction using 1990CTL, the other control simulations are also bias-corrected.

**Code availability**. The code that supports the findings of this study is available from the corresponding author on request.

## Data availability
The source code of the climate model can be found at https://www.gfdl.noaa.gov/cm2-5-and-flor. The data that support the findings of this study are available from the corresponding author on request.

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

## Acknowledgements

Kieran Bhatia and Gabriel Vecchi were supported by National Science Foundation under Grant AGS-1262099 and in part by the Carbon Mitigation Initiative at Princeton University BP International 02085(7)" to "Kieran Bhatia and Gabriel Vecchi were supported by National Science Foundation under Grant NSF EAR-1520683 and in part by the Carbon Mitigation Initiative at Princeton University BP International 02085(7). Any opinions, findings, and conclusions or recommendations expressed in this material are those of the authors and do not necessarily reflect the views of the National Science Foundation.

## Author contributions

K.B. designed the study, completed a majority of the analysis, and wrote the entirety of the manuscript. G.V. helped develop the HiFLOR model, and G.V. and H.M carried out the experiments. G.V. also provided key suggestions on research topics to explore. H.M. provided key suggestions on the analysis. T.K. assisted in the statistical analysis and provided guidance on the analysis. J.K. developed ADT-HURSAT data and provided it for analysis. J.K. provided minor suggestions on experimental design. K.D. helped design and write a description of the bias corrections for HiFLOR. C.W. was the primary developer of the bias corrections for HiFLOR.

## Additional information

**Competing interests:** The authors declare no competing interests.

