## [Peer Review File · Nature Communications]

Reviewer #1 (Remarks to the Author):

Review of Rhatia et al.

General comments: I have to be perfectly honest, with several of my very respected authors on the list of contributors, this manuscript is unfortunately not living up to the standard of logical scrutiny:

(1) The authors well acknowledged the deficiencies in the two dataset for properly detecting global intensity changes given inhomogeneity in the intensity track methods across different basins, the availability and resolution of satellites in different regions. It is disturbing to see the correlations of the two global datasets is only 0.13 that includes the high-correlated areas of the Atlantic basin (Supplemental Figure 2) which implies that there is literally no or even negative correlation if the Atlantic basic is excluded from the dataset. In other words, to claim global increase in intensification is a gross overstatement, and shall not have been included in the lengthy discussion in the first place.

(2) As also acknowledged by the authors, given the ADT-HURSAT is trained more to fit the Atlantic basin, and given it uses the IBTrACS already used a significant amount of ADT-related techniques especially when airborne in-situ observations are not available, it is no surprising that the two datasets are highly correlated. Even with that in mind, the difference in the range of interests to this study at positive meaningful detectable intensification ranges ($>10\text{knt}/24\text{h}$) (supplementary figure 1b) is probably much larger than any detectable trend claimed here.

(3) Further adds to the uncertainty of attribution is the acknowledged likely contribution of multi-decadal oscillation to the the uptick in TC intensity towards the later part of the time series. Obviously overall stronger storms will have on average stronger intensification. Again, the statement on the evidence of anthropogenic contribution to the trend is not adequately justified from observations in this reviewer's opinion.

(4) It further raises concern to this reviewer that the original HiFLOR simulated intensity changes in Figure 4b for the Atlantic basin is too far from those of the two observational datasets, especially for the stronger intensity rates that are the focus of the study. Any bias correction using the observed datasets will become dependent on the observational datasets and thus cannot be used as an additional piece of evidence to support that the model is consistent with the observed change in trend.

For the aforementioned reasons, I regrettable cannot recommend its acceptance to this prestigious journal.

Reviewer #2 (Remarks to the Author):

In this work, the authors take a multipronged approach to investigating changes in the rate of intensification in tropical cyclones over time and in different climate regimes—rapidly intensifying tropical cyclones are a large source of forecast error, which can have a significant human impact. To investigate the trends, the authors leverage two observational datasets of tropical cyclone intensity (one subjective and another objective) and bias-corrected climate model output to investigate changes to intensification rates.

Despite being the best-available, long-term data, the datasets used by the authors are not without issues—the IBTrACS dataset is inconsistent between different forecast centers and the ADT-HURSAT misses intensity trends. However, I believe that the authors adequately address the potential pitfalls and limitations associated with the observations (e.g., excluding the Indian Ocean for not having a sufficient period of record) to illustrate an increase in rapid intensification rates over the 28-year period covered by the datasets.

In addition to observations, the authors use a novel approach to bias correct climate model output with the first 17-years of the observational datasets. Then, they examine trends under various climate runs. The simulations reinforce the finding that there is an upward trend in intensification rates. Given that rapid intensification is a challenging element to tropical cyclone forecasting, I believe that the findings of this work are important from both a scientific and societal standpoint.

Overall, I feel that the authors provide a clearly written, well-constructed study for demonstrating an uptick in the rate of tropical cyclone intensification and I recommend that this manuscript be accepted with minor revisions—most of my comments regard clarifications that should have little impact of the results or conclusions of this work.

Major comment(s):

I am a bit confused by the text discussing Supplementary Figure 3 and the caption for the figure. I believe that the 2008 in lines 223 and 225 should be 1998 to represent the 17-year period centered on 1990. If the year is 1998, then the change in the tails of the probability densities between Fig. 1 and Sup. Fig. 3 makes sense. If only the year 2009 is excluded from Supplementary Figure 3, I have some concerns as to why the tails of the IBTrACS and ADT-HURSAT distributions drop by 1-3 percentage points. Related to this comment, there are a few other places in the manuscript where I think that the 2008 either needs to be a 2009 or 1998 (e.g., line 272). I believe the majority of these are typos. But, I ask that the authors take time to check all their years to ensure that the ranges are appropriate to assist the reader in interpreting the authors' work.

Figure 6 is telling visual for this work. However, the authors do not include a discussion in the methods section. Given the criticisms over this type of plot (e.g., Wilks 2016), I would like to see a further discussion to see if the authors took steps to ensure that a stricter standard for statistical significance was taken given that ~450 realizations of the binomial proportion statistic are being examined in each panel. If the authors did not consider field significance, I'd be interested to hear how many grid points would pass with a simple constraint to multiple hypothesis testing like the Walker's criteria under the generous assumption that the grid boxes are statistically independent. While I am interested in this metric, the authors' finding that "the simulations are too idealized to strictly attribute the identified growth in RI to anthropogenic forcing" will remain the same regardless of the extra statistical rigor.

Minor comment(s):

On Page 6 Lines 139 to 140, the authors state that "ADT-HURSAT's shelf-like pattern suggests that RI is a special process that could preferentially augment the probability of the highest intensification rates." While the claim that RI is a special process may hold merit, the authors' argument seems inconsistent with their statement page 15 lines 342 to 353 in the methods section regarding the limitations of the ADT-HURSAT. I feel that the shelf-like pattern is a result of ADT-HURSAT catching up rather than capturing a unique quality of RI. Anyway, I consider this a minor comment because it relates more to the authors interpretation of the shelf-like pattern and, to my understanding, would not impact on the findings.

Typos:

I only noticed a few places where hyphens (e.g., best-track) and en dashes (e.g., Saffir-Simpson) are needed. These typos are minor, and I assume that these changes will be addressed by a copy editor so I am not including line numbers.

References:

Wilks, D. S., 2016: "The stippling shows statistically significant grid points": How research results are routinely overstated and overinterpreted, what to do about it. *Bull. Amer. Meteorol. Soc.*, 97, 2263–2273.

Response to Reviewers

Reviewer #1

(1) The authors well acknowledged the deficiencies in the two dataset for properly detecting global intensity changes given inhomogeneity in the intensity track methods across different basins, the availability and resolution of satellites in different regions. It is disturbing to see the correlations of the two global datasets is only 0.13 that includes the high-correlated areas of the Atlantic basin (Supplemental Figure 2) which implies that there is literally no or even negative correlation if the Atlantic basic is excluded from the dataset. In other words, to claim global increase in intensification is a gross overstatement, and shall not have been included in the lengthy discussion in the first place.

Thank you for highlighting the lack of correlation globally between IBTrACS and ADT-HURSAT. We admit that the inclusion of the IBTrACS results without strongly caveating them is potentially distracting to the readers of this manuscript and therefore must be remedied. As a result, we have removed and significantly toned down the conclusions relating to the global trends. Instead, we focus more on the Atlantic basin results and emphasize the need for a longer higher quality homogenized dataset for future studies to properly evaluate global trends. We have adjusted the abstract, results, and conclusion to reflect these changes. As a result, we have made it clear that:

- 1) We don't trust IBTRaCS global trend results, which disagree with ADT-HURSAT.
- 2) ADT-HURSAT results are not yet significant globally. A longer dataset might lead to a significant result based on the unusually high trend.

This is consistent with Kossin et al. (2013), which found that the trends in IBTrACS intensity are spuriously inflated everywhere, but least so in the Atlantic basin. This comes as no surprise given the high level of reanalysis in the Atlantic compared to other basins, and the more consistent operational protocols at the National Hurricane Center. This is also consistent with the heuristic exercise of Kossin et al. (2013), which showed that the observed changes in the environment would not and should not yet support a detectable trend in intensity. That is, the emergence timescale is longer than the data period used here.

As the manuscript is substantially revised, we will not list all the quoted modifications here but instead, we will highlight some major ones.

- A. Our abstract and discussion sections completely avoid any mention of the significance and direction of the global trend.

- B. We removed this paragraph, “Based on the physical mechanisms proposed in previous studies¹³⁻¹⁵, the large increase in the probability of the most extreme intensity changes captured by IBTrACS is plausible. It is possible that resolution and algorithm issues with ADT-HURSAT prevent it from resolving the intensity changes that have displayed the largest trends. On the other hand, the global influx of new satellites has enabled better detection of intensity changes in the later years and could artificially enhance the slope of the extreme quantiles in IBTrACS. Further investigation determining which observational dataset is closer to the truth is absolutely necessary because of the important ramifications for the safety of coastal areas around the world.” This text confused the readers and the important conclusions of our manuscript.
- C. We make it clear in our discussion of global trends that ADT-HURSAT is the homogenized dataset and therefore, it is the only one that should be trusted. Please this restructured paragraph below from our results section, “For the global data, ADT-HURSAT trends are clearly much smaller than those in IBTrACS, and since IBTrACS trends are likely more susceptible to data homogeneity problems, we conclude that the data support only the weak global trends shown by ADT-HURSAT. This conclusion is particularly notable because multiple studies^{4,13-15} have assumed that IBTrACS is reliable during the time frame considered in our study when in reality, there is likely a spurious trend masked by the data. It is possible that resolution and algorithm issues with ADT-HURSAT prevent it from resolving the intensity changes and have slightly dampened largest trends. However, it is much more likely that the influx of new satellites has enabled better detection of intensity changes in the later years of our time series and thus, artificially enhanced the slope of the extreme quantiles in IBTrACS.”

Now that we have demonstrated (with additional examples scattered throughout the manuscript) that we have heavily caveated the global results and placed the emphasis on the ADT- HURSAT results, we also want to mention why we do not completely remove IBTrACS and global data from the discussion altogether. As mentioned in the text, multiple studies (see references 4, 13-15 for just a few) use IBTrACS to make sweeping conclusions about TC trends and behavior with little discussion of the uncertainty that underlies this dataset. However, we show with the ADT-HURSAT and Atlantic basin results that IBTrACS strongly inflates intensity-change based trends globally. No previous manuscript has demonstrated this conclusion and therefore, we include these results in our manuscript.

(2) As also acknowledged by the authors, given the ADT-HURSAT is trained more to fit the Atlantic basin, and given it uses the IBTrACS already used a significant amount of ADT-related techniques especially when airborne in-situ observations are not available, it is no surprising that the two datasets are highly correlated. Even with that in mind, the difference in the range of interests to this study at positive meaningful detectable intensification ranges (>10knt/24h) (supplementary figure 1b) is probably much larger than any detectable trend claimed here.

It is very difficult to understand what the reviewer is referencing in this comment, but we will try to address one potential interpretation. ADT-HURSAT is used as the primary source of intensity estimates in basins other than the Atlantic basin. We have removed the text, “Additionally, there is evidence that the ADT-HURSAT algorithm was more carefully trained in the Atlantic basin¹⁵” from the manuscript to avoid confusing the readers. Therefore, the reviewer’s comments about it being ‘no surprise’ is not fair especially because no previous manuscript has provided this intensity change comparison.

The reviewer’s comments regarding the “difference in the range on interests” is also very confusing. The differences in IBTrACS and ADT-HURSAT intensity change distributions is not relevant to the detectable trend in the ADT-HURSAT data. For our purpose, we just want to see if the trend is large enough to stand out from natural variability and therefore is detectable.

Furthermore, we now emphasize in the text that ADT- HURSAT is the only reliable dataset for trend analysis. IBTrACS likely has lower intensity errors when looking at individual storms and seasons but its differences among basins and seasons make its trend results highly questionable. We still show figure 1 and supplementary figure 1 to emphasize that even with the coarsening imposed on ADT-HURSAT data to ensure it is homogeneous, ADT-HURSAT closely resembles IBTrACS. Therefore, ADT-HURSAT has physically-justified intensity changes that just have slightly higher errors than IBTRaCS. Therefore, ADT-HURSAT is the only available choice for trend analysis because it also has no systematic biases temporally. Therefore, the cited correlation comments with IBTrACS does not detract from the ADT-HURSAT results.

(3) Further adds to the uncertainty of attribution is the acknowledged likely contribution of multi-decadal oscillation to the uptick in TC intensity towards the later part of the time series. Obviously overall stronger storms will have on average stronger intensification. Again, the statement on the evidence of anthropogenic contribution to the trend is not adequately justified from observations in this reviewer's opinion.

We thank the reviewer for this comment, and we have revised the detection and attribution statements in the paper. The detection and attribution statements are further caveated, and the resulting uncertainty is stated. Please see the abstract for a great summary of our new focused attribution statements, “Our results suggest a detectable increase of Atlantic intensification rates with a positive contribution from anthropogenic forcing and reveal a need for more reliable data before detecting a robust trend at the global scale. “

A great example of these revisions is visible in the paragraph before the final discussion (figure 6 nicely justifies these claims),

“Increased greenhouse gas forcing in HiFLOR clearly leads to more frequent large intensification rates, but the available simulations are too idealized to strictly attribute the identified growth in RI during the period 1982 to 2009 to anthropogenic forcing. An ensemble of experiments that simulate anthropogenically-forced climate change during

this specific period, including changes in RI ratio, would be better suited for more comprehensive attribution analysis. Thus, for now, we can only conclude that anthropogenic forcing significant increases in extreme TC intensification rates in the HiFLOR model compared to pre-industrial (1860CTL) conditions.”

Please see our last paragraphs as an example of our awareness of the caveats associated with our results. This paragraph also explains why we have provided careful and justified statements regarding the evidence of an anthropogenic contribution,

“By itself, a 28-year upward trend in a TC intensification metric does not necessarily reflect the effects of anthropogenic climate forcing because of the intrinsic natural variability of the climate system. Here, we use bias-corrected TC intensification rates simulated by HiFLOR to demonstrate that natural variability cannot explain the magnitude of the observed upward trend in the Atlantic basin. These conclusions are possible because HiFLOR is a unique climate model that can successfully simulate the most intense TCs and highest intensification rates in multicentury simulations.

This study is limited by the ability of a climate model to accurately represent natural variability as well as the uncertainty around the trends in the observational records for a short period. However, this study represents a crucial first step in quantifying the precise roles of stochastic processes, anthropogenic warming, and natural variability when assessing changes in TC intensification rates. Further analysis with additional high-resolution climate models and a longer and more reliable observational record is required to confirm these conclusions. Regardless, these trends provide another example of the potentially serious repercussions of anthropogenic warming for TCs.”

Finally, we would like to be clear that we account for natural variability (multi-decadal oscillations) with the model, HiFLOR, that is able to accurately capture climate variations. The comment about ‘stronger storms will have on average stronger intensification’ is true but the magnitude of the changes has not been attributed carefully before. Our model framework allows us to diagnose normal climate variability and differentiate it from the observed trends.

(4) It further raises concern to this reviewer that the original HiFLOR simulated intensity changes in Figure 4b for the Atlantic basin is too far from those of the two observational datasets, especially for the stronger intensity rates that are the focus of the study. Any bias correction using the observed datasets will become dependent on the observational datasets and thus cannot be used as an additional piece of evidence to support that the model is consistent with the observed change in trend.

The HiFLOR model prior to bias correction has too little variance. We have added text to the manuscript to explain the importance of the bias correction, “When we explore the potential influence of anthropogenic forcing on TC intensification rates, it is critical that HiFLOR provides a realistic picture of natural climate variability. Bias-corrections increase the probability of extreme intensity changes in the HiFLOR control simulations, which broadens the intensity change distribution. Thus, the bias-corrections bolster the year-to-year variations in RI ratio and the range of the RI ratio slopes in the HiFLOR

simulations, which strengthens the statistical tests for establishing significant observational trends.”

Note that we also include in our methods section, “The formulation of QDM aims to preserve relative changes in model-simulated climate variable quantiles. In other words, with respect to the quantiles, QDM guards against distorting the underlying climate model’s climate sensitivity.”

Finally, as a check, we have recreated Fig. 5 but for non-bias-adjusted data. The results are similar but more significant because of the reduced RI ratio variance. We do not want it to be “too easy” for an observed trend to be detectable (outside range of model variability). So, by bias adjusting, we are trying to make our analysis more conservative, with a higher bar for detectable signal. We hope the reviewer understands the importance of this step.

Reviewer #2:

Major comment(s):

I am a bit confused by the text discussing Supplementary Figure 3 and the caption for the figure. I believe that the 2008 in lines 223 and 225 should be 1998 to represent the 17-year period centered on 1990. If the year is 1998, then the change in the tails of the probability densities between Fig. 1 and Sup. Fig. 3 makes sense. If only the year 2009 is excluded from Supplementary Figure 3, I have some concerns as to why the tails of the IBTrACS and ADT-HURSAT distributions drop by 1-3 percentage points. Related to this comment, there are a few other places in the manuscript where I think that the 2008 either needs to be a 2009 or 1998 (e.g., line 272). I believe the majority of these are typos. But, I ask that the authors take time to check all their years to ensure that the ranges are appropriate to assist the reader in interpreting the authors' work.

We thank the reviewer because these typos significantly change the interpretation of this paper. We have changed lines 223 and 225 to 1998. Line 272 has also been fixed. 2008 is now no longer mentioned in the paper as an end date; the previous inclusions were all typos.

Figure 6 is telling visual for this work. However, the authors do not include a discussion in the methods section. Given the criticisms over this type of plot (e.g., Wilks 2016), I would like to see a further discussion to see if the authors took steps to ensure that a stricter standard for statistical significance was taken given that ~450 realizations of the binomial proportion statistic are being examined in each panel. If the authors did not consider field significance, I'd be interested to hear how many grid points would pass with a simple constraint to multiple hypothesis testing like the Walker's criteria under the generous assumption that the grid boxes are statistically independent. While I am interested in this metric, the authors' finding that "the simulations are too idealized to strictly attribute the identified growth in RI to anthropogenic forcing" will remain the same regardless of the extra statistical rigor.

Thank you for pointing out the lack of a discussion in the methods section. This text was added to the uncertainty quantification section of the Methods section:

"For the creation of Figure 6, random error is not added to the data because we only compare spatial differences in RI ratio among the different HiFLOR simulations. Statistical significance is computed using a binomial proportion test with p values below 0.05 considered significant³⁷. Data is only plotted in a grid box if there is at least $\frac{1}{4}$ of a TC day per year in the two HiFLOR simulations used to calculate the percent difference."

Regarding the comment about the Walker Criteria, we have looked over the referenced text and are confused where the "450 realizations" comments comes from. We assume that you are calculating the total number of grid boxes in the active basins. If we plug this value into equation 2 from (Wilks, 2016), we calculate the significance test changes

from a p value of 0.05 to 0.00013. We feel this test is too strict and roughly cuts our significant grid boxes in half. As is, we use a well-cited significance test measure and mention the caveats of our results. Therefore, we thank the reviewer for this suggestion, but we feel that the inclusion of the Walker criteria is not essential.

Note also that we list the percent of area that passes the local significance test for each map in Fig. 6. This is 41 grid boxes (9.7 % of all shaded gridboxes) in 1940CTL, 156 (34.7% of all shaded grid boxes) in 1990 CTL, and 164 (38.2% of all shaded gridboxes) in 2015CT. While we have not performed a Monte Carlo test following Livezey and Chen (1983) to establish field significance, the large fraction of local tests with statistical significance strongly suggest that the projections for 1990CTL and 2015CTL have field significance.

References

Livezey, R.E. and W.Y. Chen, 1983: Statistical Field Significance and its Determination by Monte Carlo Techniques. *Mon. Wea. Rev.*, **111**, 46–59.

Minor comment(s):

On Page 6 Lines 139 to 140, the authors state that “ADT-HURSAT’s shelf-like pattern suggests that RI is a special process that could preferentially augment the probability of the highest intensification rates.” While the claim that RI is a special process may hold merit, the authors’ argument seems inconsistent with their statement page 15 lines 342 to 353 in the methods section regarding the limitations of the ADT-HURSAT. I feel that the shelf-like pattern is a result of ADT-HURSAT catching up rather than capturing a unique quality of RI. Anyway, I consider this a minor comment because it relates more to the authors interpretation of the shelf-like pattern and, to my understanding, would not impact on the findings.

Thank you for this suggestion. We added, “The former interpretation is much more likely considering the ADT-HURSAT algorithm has documented problems during TC eye formation (see Methods).”

Typos:

I only noticed a few places where hyphens (e.g., best-track) and en dashes (e.g., Saffir-Simpson) are needed. These typos are minor, and I assume that these changes will be addressed by a copy editor so I am not including line numbers.

Thank you. We have made these edits.

Reviewer #1 (Remarks to the Author):

The authors have addressed all my previous round of review comments adequately, substantially reduced the scope and the overstatement of the findings. With a much reduced scope and shorter statistics over only the Atlantic basin, the significance of the findings and the direct connection of the trend with the warming climate or nature variations are questionable, though it is still a valuable study.

Response to Referees

Reviewer 1 has no specific suggestions to improve the manuscript, and we did not receive any feedback from Reviewer 2 in this round.